# A Review on the Adoption of AI, BC, and IoT in Sustainability Research

Susie Ruqun WU [1], Gabriela Shirkey [2,3,*], Ilke Celik [4], Changliang Shao [5] and Jiquan Chen [2,3]

1   SusDatability Ltd., A2205 Chuangxin Plaza, Pinshan Ave. 2007, Pinshan District, Shenzhen 518118, China; sw@susdatability.com
2   Center for Global Change and Earth Observations, Michigan State University, 218 Manly Miles Building, 1405 S. Harrison Road, East Lansing, MI 48823, USA; jqchen@msu.edu
3   Department of Geography, Environment and Spatial Sciences, Michigan State University, 673 Auditorium Rd., East Lansing, MI 48824, USA
4   Civil and Environmental Engineering (CEE), South Dakota School of Mines & Technology, 501 E. Saint Joseph St., Rapid City, SD 57701, USA; ilke.celik@sdsmt.edu
5   Institute of Agricultural Resources and Regional Planning, Chinese Academy of Agricultural Sciences, No. 12 Zhongguancun South St., Haidian District, Beijing 100081, China; shaochangliang@caas.cn
*   Correspondence: shirkeyg@msu.edu

**Abstract:** The rise of artificial intelligence (AI), blockchain (BC), and the internet of things (IoT) has had significant applications in the advancement of sustainability research. This review examines how these digital transformations drive natural and human systems, as well as which industry sectors have been applying them to advance sustainability. We adopted qualitative research methods, including a bibliometric analysis, in which we screened 960 publications to identify the leading sectors that apply AI/BC/IoT, and a content analysis to identify how each sector uses AI/BC/IoT to advance sustainability. We identified "smart city", "energy system", and "supply chain" as key leading sectors. Of these technologies, IoT received the most real-world applications in the "smart city" sector under the dimensions of "smart environment" and "smart mobility" and provided applications resolving energy consumption in the "energy system" sector. AI effectively resolved scheduling, prediction, and monitoring for both the "smart city" and "energy system" sectors. BC remained highly theoretical for "supply chain", with limited applications. The technological integration of AI and IoT is a research trend for the "smart city" and "energy system" sectors, while BC and IoT is proposed for the "supply chain". We observed a surge in AI/BC/IoT sustainability research since 2016 and a new research trend—technological integration—since 2020. Collectively, six of the United Nation's seventeen sustainable development goals (i.e., 6, 7, 9, 11, 12, 13) have been the most widely involved with these technologies.

**Keywords:** literature review; sustainability; AI; BC; IoT; smart city; energy system; supply chain

## 1. Introduction

Many challenges facing our planet today, such as environmental degradation, disaster relief, and climate change, are due to failed sustainable socioecological transformation. A "worldwide growth in affluence" has been identified as the root cause of the challenges that threaten natural systems, economies, and societies by some scholars, creating a call for the reassessment of "growth-oriented economies" [1]. Others seek to ultimately transform the way we live on (or possibly beyond) Earth via breakthrough technologies. We, on the other hand, seek a socioecological transition (e.g., adjustments affecting dynamics between societal energy regimes and codependent ecological changes) that steers us toward sustainability under the current development path (e.g., economic growth coupled with environmental problems) and employs emerging technological advancements.

Among the most important technological advancements, artificial intelligence (AI), blockchain (BC), and the internet of things (IoT) are expected to be the critical drivers of

digital transformation in the upcoming decades [2], which will "rewire our future" [3] and have revolutionary impacts on nature and human society. At the fifth session of the United Nations Environment Assembly in 2021, digital transformation was identified as a key part of the UN Environment Programme's path toward its sustainable development goals (SDGs) [3]. Applications are constantly evolving to unlock the power of digital technology, ultimately creating "an inclusive, human-centered future" [4]. It is now a pivotal moment in the history of environmental science to bring together new technologies to create knowledge and insight at local and global scales [5].

AI and algorithms capable of performing tasks that typically require human intelligence have rapidly changed the way people interact with each other and with the environment, and they are expected to produce long-lasting impacts on social, economic, and environmental sustainability, both in short- and long-term scales [6]. Previously, due to a lack of data, 68% of the 93 SDG indicators on the environmental dimensions of sustainability could not be tracked [7]. However, new opportunities afforded by big data could tackle these obstacles, as they provide environmental insight in near real-time [5]. For example, AI and big data can help optimize energy system demand and supply modeling, enable remote work platforms to mobilize the contingent workforce, and provide a simulation framework for animal, plant, and habitat interactions [4].

Blockchain (BC) and the building blocks of this revolutionary technology (e.g., decentralization, trust-building, third-party engagement, consensus-based participation) stimulate democracy and highly coincide with participatory approaches to reach sustainability. BC has been seen as the prime solution for developing a smart and circular economy [8]. Global giants in the food business are adopting BC to ensure traceability [9] and more reliable, effective, fuel-efficient, and safe transportation management can be realized through BC [10]. Wang and Su (2020) foresee the possibility of integrating BC and distributed/decentralized energy holds in the future [11]. BC also has the potential to become a game-changer for audit processes because it is inherently resistant to the modification of stored data [12], which could engender a radical shift in sustainability assessments, such as carbon emissions tracking, carbon trading, environmental/social performance tracking, and verification.

Air and water quality, energy management, waste treatment, infrastructure, transportation, and pandemic preparedness and response are major environmental issues facing most cities around the world. Here, IoT offers excellent potential for building smart cities through game-changing innovations in information and communications technology (ICT). Network sensors connected to energy-consuming devices can communicate with utility services to balance power generation and energy usage and prevent crashes or unexpected outages [13]. For example, New York Waterway ferries aggregate data from all connected sensors to a central dashboard in real-time and apply predictive and prescriptive analytics to support intelligent public transportation systems [14]. Siemens launched a City Air Management solution that displays real-time air quality data detected by sensors across a city and predicts values for the upcoming three to five days [15]. Bigbelly builds on IoT and its cloud platform to provide innovative waste and recycling solutions for public spaces [16].

However, these technological innovations are not without challenges, which raise new research directions to improve productivity and performance (e.g., smart factory and manufacturing), as well as smart cities and sustainable growth. For example, Industry 4.0, also known as the fourth industrial revolution, is a large consumer of energy in the end-user sector and generates significant GHG emissions. Further challenges include energy consumption, network reach, and green manufacturing, as well as energy peak loads, costs, and balance. These challenges are active areas of research for AI and IoT applications in an effort to monitor and predict energy patterns and emissions [17].

With the prospect that these new technologies might fully bloom in the upcoming decades across multiple scientific disciplines, the question remains how, whether standalone or integrated, they can help meet the SDGs. Focusing on the aforementioned

technologies, our study objective is to examine the potential adoption of these technologies for sustainability research via literature review. Specifically, we endeavor to answer the following two research questions (RQs): (1) What venues are addressing the research focus (i.e., application of AI/BC/IoT in the context of sustainability), what are the research trends, and what are the leading industries/sectors adopting AI/BC/IoT? (2) How are the leading sectors addressing AI/BC/IoT or adopting them in real-world applications to promote sustainability? While some adverse effects of these new technologies (e.g., social unrest, carbon emissions from crypto currency, energy consumption) are debated [18], an assessment of the technologies themselves is outside the scope of this study.

## 2. Materials and Methods

To achieve our objectives, a systematic literature review was conducted with keywords, applying Boolean operators that searched across article titles, abstracts, and keywords (TAK) in a major academic database—the Web of Science (WoS) Core Collection. The search was conducted in January 2021, with an initial sample of 1433 publications. WoS was selected because it covers a wide range of multidisciplinary publications and is well suited to evidence synthesis in the form of systematic reviews [19]. The Bibliometrix R-package [20] was used for WoS publication bibliometric analysis.

To answer RQ1, the following workflow was performed in order (Figure 1):

A.  Identification of the search string to perform the initial search on TAK.
B.  Screening the initial publications to exclude the following types of publications:

- Those appearing on TAK search results without any focus on AI/BC/IoT, e.g., agricultural intensity and appreciative inquiry—the same "AI" initials for artificial intelligence;
- Those solely focused on the technical aspects of the technology itself without any relevance to sustainability, e.g., LoRa (long range) technique, Message Queuing Telemetry Transport (MQTT) protocol, mesh networking, networking design, IoT architecture/design;
- Those discussing and/or assessing the sustainability of the technology itself, such as the energy consumption of technologies (e.g., cryptocurrency mining), the energy harvesting of IoT devices, social justice, law and ethics, security and privacy issues of AI/BC/IoT (while important, this cohort of studies needed to be separated, as the scope and focus of this study is to analyze the application of AI/BC/IoT rather than placing them as the studied objects);
- Those solely discussing the technologies within a specific industry (e.g., fintech, accounting, banking, real estate, dentistry, fine arts, linguistic, radiology, music recording industry) without any relevance to sustainability under the scope of this study (i.e., to improve quality of life, the efficiency of urban operation and services, concerning economic, social, environmental as well as cultural aspects) [21].

C.  Performance of the bibliometric analysis on the screened sample, including:

- Total annual scientific production to observe the changing research interest in the subject;
- Analysis of the most relevant sources and the collaboration network of authors' institutions to reveal the highest contributing venues;
- Analysis of the word dynamics and trend topics using authors' keywords by counting yearly occurrences of top keywords to identify leading sectors, followed by a cooccurrence network—visualizing the conceptual structure in a two-dimensional plot through the interconnection of terms within the TAK—to recognize the most recurrent themes [20]. Because the frequency of the keywords impacts the cooccurrence map (the lower the term frequency, the more complex and less readable the network), we constructed the map using keywords recurring at least three times as the best possible tradeoff [22].

D.     Performance of a subsequent literature search for each leading sector identified using the search string formulated for each sector (Table 1). The samples obtained were then intersected with the initial sample from Step B to ensure exclusion of irrelevant studies.

E.     Review of the studies from Step D and their respective TAK to enhance validity and ensure their relevance to each sector.

F.     Bibliometric analysis on each leading sector, using a cooccurrence network, to identify clusters of research interests and key technologies adopted.

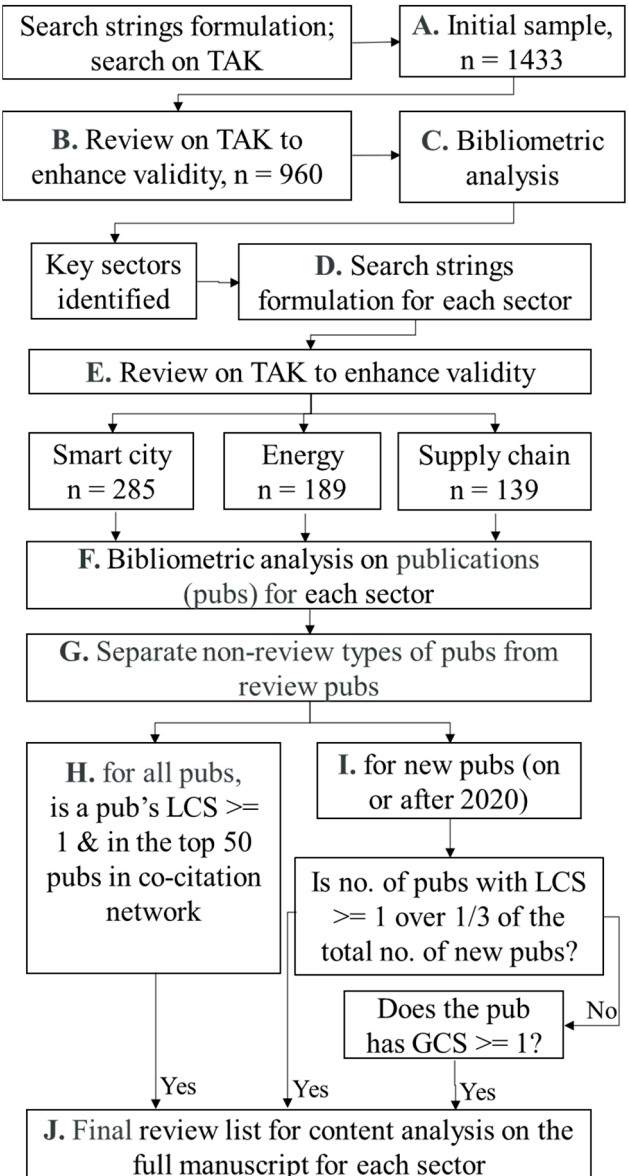

**Figure 1.** The overall research workflow, literature search, and analysis to answer RQ1 and RQ2. "N" indicates the number of samples initially obtained/screened/reviewed. TAK—title, abstract, keywords; pub—publication; LCS—local citation score; GCS—global citation score.

**Table 1.** Search strings formulated for title, abstract, and keywords (TAK) search and the search results, including total number of publications from initial sample and number of publications after screening.

| Search Strings | Sector | Initial Sample | After Screening |
|---|---|---|---|
| (Sustainability OR "sustainable development") AND (blockchain OR "internet of things" OR "IoT" OR "AI" OR "artificial intelligence") | N/A | 1433 | 960 |
| (Sustainability OR "sustainable development") AND (city OR cities OR "smart cities" OR "smart building") AND (blockchain OR "internet of things" OR "IoT" OR "AI" OR "artificial intelligence") | Smart cities | 444 | 285 |
| (Sustainability OR "sustainable development") AND (energy OR "smart grid" OR "energy management" OR "energy efficiency" OR "renewable energy") AND (Blockchain OR "internet of things" OR "IoT" OR "AI" OR "artificial intelligence") | Energy | 442 | 189 |
| (Sustainability OR "sustainable development") AND ("supply chain" OR "supply chain management" OR "logistics" OR "procurement" OR "traceability") AND (Blockchain OR "internet of things" OR "IoT" OR "AI" OR "artificial intelligence") | Supply chain | 192 | 139 |

When conducting research flow E, a further intersection with the initially screened 960 publications was necessary even though the search strings included both "sustainability" and the sector-specific terms. For example, without performing an intersection, the most locally cited paper among the initial sample for the energy sector was "Sustainability of bitcoin and blockchains (https://doi.org/10.1016/j.cosust.2017.04.011 (accessed on 27 January 2021))", which was irrelevant to the scope of this study and was initially screened out.

To answer RQ2, a content analysis was performed on a further cohort of selected key publications, acquired through the following steps:

G.  Review articles were selected first for each sector to: (1) acquire a general understanding of the topic and (2) keep non-review articles for further content analysis.
H.  The list of key publications for each sector was identified from the union: (1) if an article's local citation score (LCS) was equal than or over 1 and (2) the top 50 publications from a historical direct citation network, ensuring that the "most relevant direct citations" of the collection [23] were included.
I.  To ensure the most up-to-date review, at least one-third of new publications (i.e., those published on or after 2020) were included in the review depending on: (1) if their LCS was equal or over 1, or (2) if the total number of publications selected from Step I.1 was less than one-third of the total new publications. Those receiving a nonzero global citation score (GCS) were added to the review list. It was quite common for new publications to receive a nonzero GCS—which considers citations from outside of the collection—while having an LCS of 0.
J.  We conducted a content analysis on the text of selected key publications, providing an overview of the research aims, solutions, AI/BC/IoT components applied, and specifically whether and how the proposed solution was applied to solve real-world problems.

There are various definitions and classifications of AI technologies [24]. Based on functionality, AI can be grouped into artificial narrow intelligence (i.e., weak AI that is trained and focused on performing specific tasks) and artificial general intelligence (i.e., strong AI that more fully replicates the autonomy of the human brain) [25,26]. Corea (2019) classified AI technologies according to problem domains and paradigms [27]. A significant subset of AI, machine learning (ML), can be classified by various mechanisms—supervised vs. unsupervised vs. reinforcement learning [28], as well as shallow vs. deep models, linear vs. nonlinear models, and more. Another term often encountered in articles about AI is "big data" or "big data analytics". Big data analytics is the use of advanced analytic techniques for "large, diverse big data sets that include structured, semi-structured and unstructured data" [26]. In this study, AI and relevant technologies are classified following definitions drawn from various sources [27,29,30], which are listed in Table 2.

**Table 2.** Classification of selected AI subsets that appeared in this review of articles, which mainly focused on machine learning.

| AI Subset | Sub-Type | | |
|---|---|---|---|
| Expert Systems | Fuzzy logic; rough set | | |
| Autonomous Systems | Robotics and intelligent systems, e.g., autonomous vehicles | | |
| Evolutionary Algorithms | Genetic algorithms (GA) | | |
| Distributed Artificial Intelligence (DAI) | Multi-agent systems (MAS); agent-based modeling (ABM); swarm intelligence | | |
| Machine learning (ML) | Decision trees (DT); | random forest; gradient boosting | |
| | Support vector machine (SVM) | | |
| ML subset: Deep learning (DL) | Artificial neural networks (ANN) | Extreme learning machine (ELM) | |
| | | Deep neural network (DNN) (with multiple hidden layers without recurrent connections) | Feedforward DNN (multilayer perception); recursive Neural networks; deep belief network (DBN); convolutional neural network (CNN) |
| | | Recurrent neural networks (RNN) (connections between units form a directed cycle) | Long short-term memory neural networks (LSTM); gated recurrent units (GRU) |

The definitions and types of IoT systems also vary widely. Himeur et al. (2020) identified three basic components: (1) IoT platform architecture (edge/fog/cloud/hybrid computing), (2) IoT technology (meters, sensors, actuators, communication strategies), and (3) IoT control algorithm [31]. Da Silva et al. (2020) further separated IoT into computing, network, and radiofrequency [17]. Lueth (2016) proposed fifteen key IoT technology components under five categories, namely, device, communication, cloud services, applications, and security [32]. Mitchell et al. (2013) proposed the Internet of Everything (IoE), which brings together four components—people, processes, data, and things—to make networked connections to enable future smart cities and communities [33]. Tran-Dang et al. (2020) categorized key IoT technologies according to function (e.g., data acquisition, connectivity, data processing) [34]. In a three-layer IoT architecture, AVSystem (2020) provided a concise definition of an IoT ecosystem: "a device collects data and sends it across the network to a platform that aggregates the data for future use by the agent" [35]. In this study, IoT and related technologies are classified following various sources [34–36], as illustrated in Figure 2.

Unlike the broad, overarching categories of AI and IoT, BC evolved from a single origin, specifically, Bitcoin. It consists of several essential layers and corresponding technologies, including: decentralized data storage, such as a distributed ledger technology (DLT); data structures, such as a Merkel tree; network protocols, such as the peer-to-peer (P2P) protocol; encryption algorithms, such as hash; consensus mechanisms, such as PoW (proof of work) or PoS (proof of stake); and smart contracts, such as a final application layer [11,37]. Access and permissions define whether a BC is public or private, and the former has the highest degree of decentralization, which allows all participants to make decisions. Wang and Su (2020) also identified alliance chains, which have a relatively higher degree of decentralization than private chains in that they allow alliance members to participate in a private chain, typically referred to as a centralized enterprise system [11]. Current mainstream platforms for building a BC solution, following Obafemi (2020), are listed in Table 3 [38].

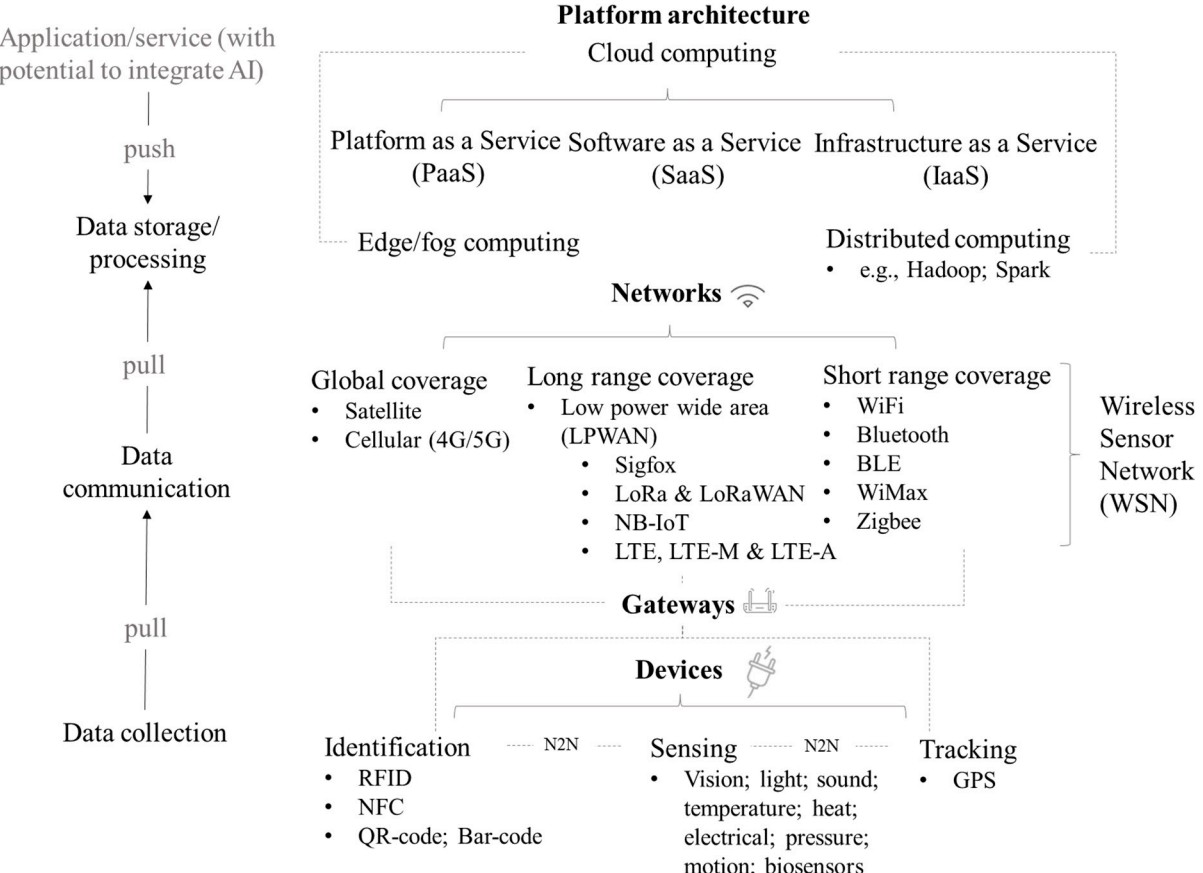

**Figure 2.** An IoT architecture and the key components engaged. These were used to identify key IoT technologies involved in the reviewed articles.

**Table 3.** Mainstream platforms for building either public or private blockchain (BC) solutions and their targeted applications [38].

| BC Platform | Type of BC | Major Application |
| --- | --- | --- |
| Bitcoin | public chain | financial transactions |
| Multichain | public chain | financial transactions |
| HyperLedger | public chain | decentralized apps (DApps) |
| EOS | public chain | DApps, smart contracts, hosting/storage solutions to blockchain projects |
| Ethereum | public chain | smart contracts |
| NEO | public chain | DApps, smart contracts, smart economy (e.g., digital identity) |
| R3 Corda | alliance/private chain | smart contracts |
| RIPPLE | alliance/private chain | connecting banks for financial transactions |

## 3. Results

### 3.1. Initial Sample

#### 3.1.1. Annual Scientific Production and Contributing Venues

The scientific community did not contribute significantly to this research topic until 2012, when the number of publications began to increase exponentially. Only 20 publications were found before 2011. In contrast, relevant publications soared from 84 in 2017 to 315 in 2020 (Figure 3a). The top four contributing venues (all scientific journals) published a total of 185 articles on the topic cumulatively (Figure 3b), accounting for almost 20% of all articles. Among the 567 total venues (including journals, conference proceedings, and

books), 21 venues issued at least four articles, making up 31% of the total, while other articles were scattered in the remaining 546 different sources.

(a)

(b)

**Figure 3.** Annual scientific production and contributing venues. (**a**) (top figure) Annual scientific production from all initially screened publications (*N* = 960) (Table S10). Relevant publications began to emerge around 2014, and a total of 144, 204, and 315 publications were observed for the years 2018, 2019, and 2020, respectively. The plot truncated at the end of 2020, and 19 publications were retrieved for the year 2021 at the time of the literature search (27 January 2021). (**b**) (bottom figure) Top 10 contributing venues (all scientific journals) up to the time of writing of this study, ordered by "h_index." "NP" is the total number of publications; "TC" is the total number of citations.

### 3.1.2. Collaboration Network

Only two clusters of research collaboration consist of at least 3 out of the top 40 contributing institutions (Figure 4a). Another two clusters only involve two engaging institutions. The remaining institutions did not collaborate with any others. The geographical location of an institution does not appear to be a factor affecting collaboration. In regard to country participation, the USA, China, and the United Kingdom lead each of the collaboration clusters (Figure 4b).

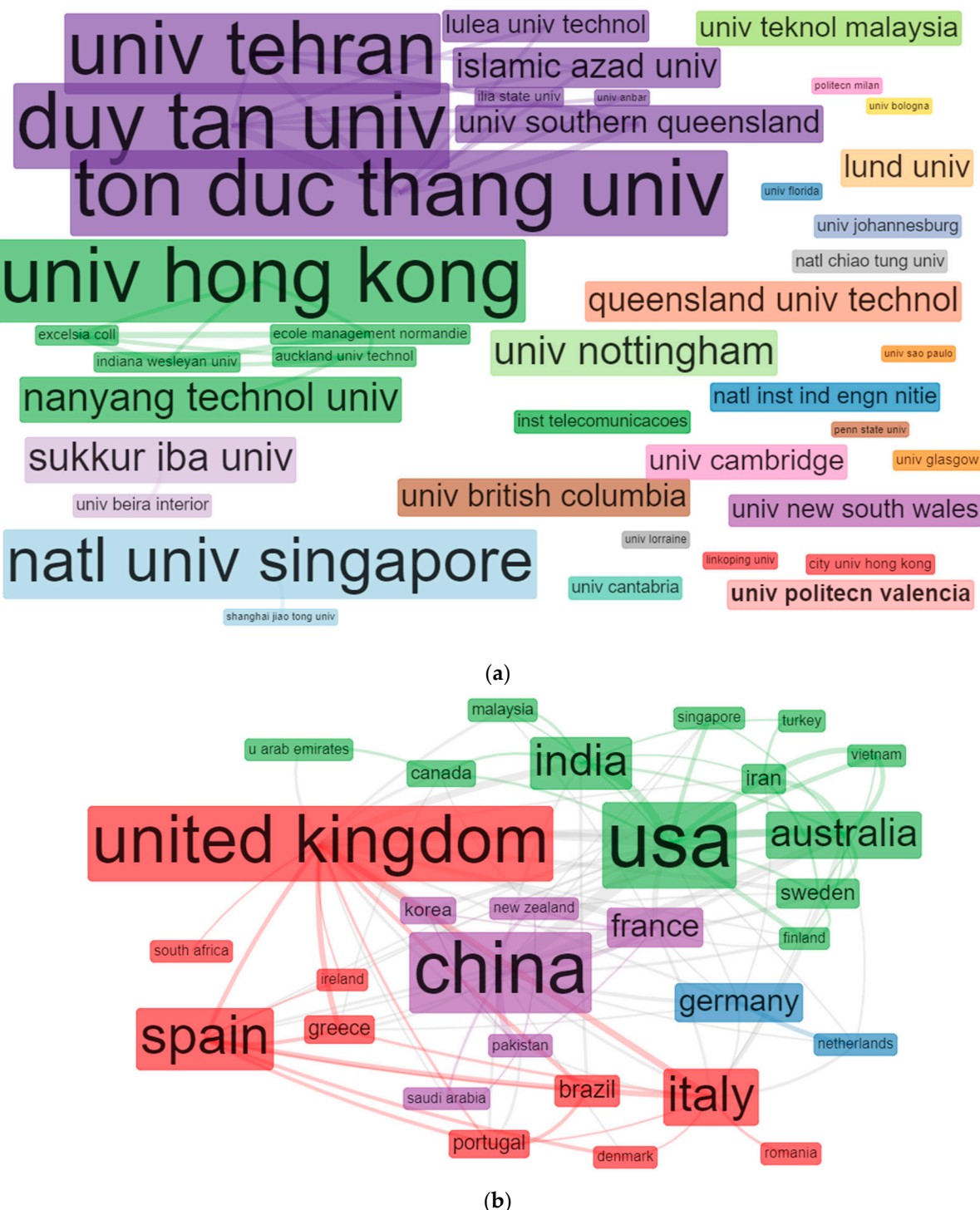

**Figure 4.** The collaboration network. (**a**) (top figure) The collaboration network between institutions from all initially screened publications (*N* = 960). Among the top 40 contributing institutions, only 18 have collaborated with one or more other institution(s), as shown in the left four clusters (purple, green, light purple, and light blue colors), while the remaining institutions are isolated nodes, indicating an overall low collaboration between institutions. (**b**) (bottom figure) The collaboration network between the top 30 contributing countries; five collaboration clusters emerge without isolated nodes.

### 3.1.3. Trend Topics and Word Dynamics

It was not until 2018 that more trend topics emerged (Figure 5a), along with more annual publications (Figure 3). Topics on smart grids and energy efficiency emerged in 2018, while 2019 added a new focus on the circular economy, with smart city/cities as a hot topic in both years. Supply chain management (SCM) emerged as a new research topic in 2020. This finding matched well with the word dynamics analysis (Figure 5b), where only smart city/cities and energy efficiency appeared among the top 10 most frequent words, with an exponential increase starting in 2014.

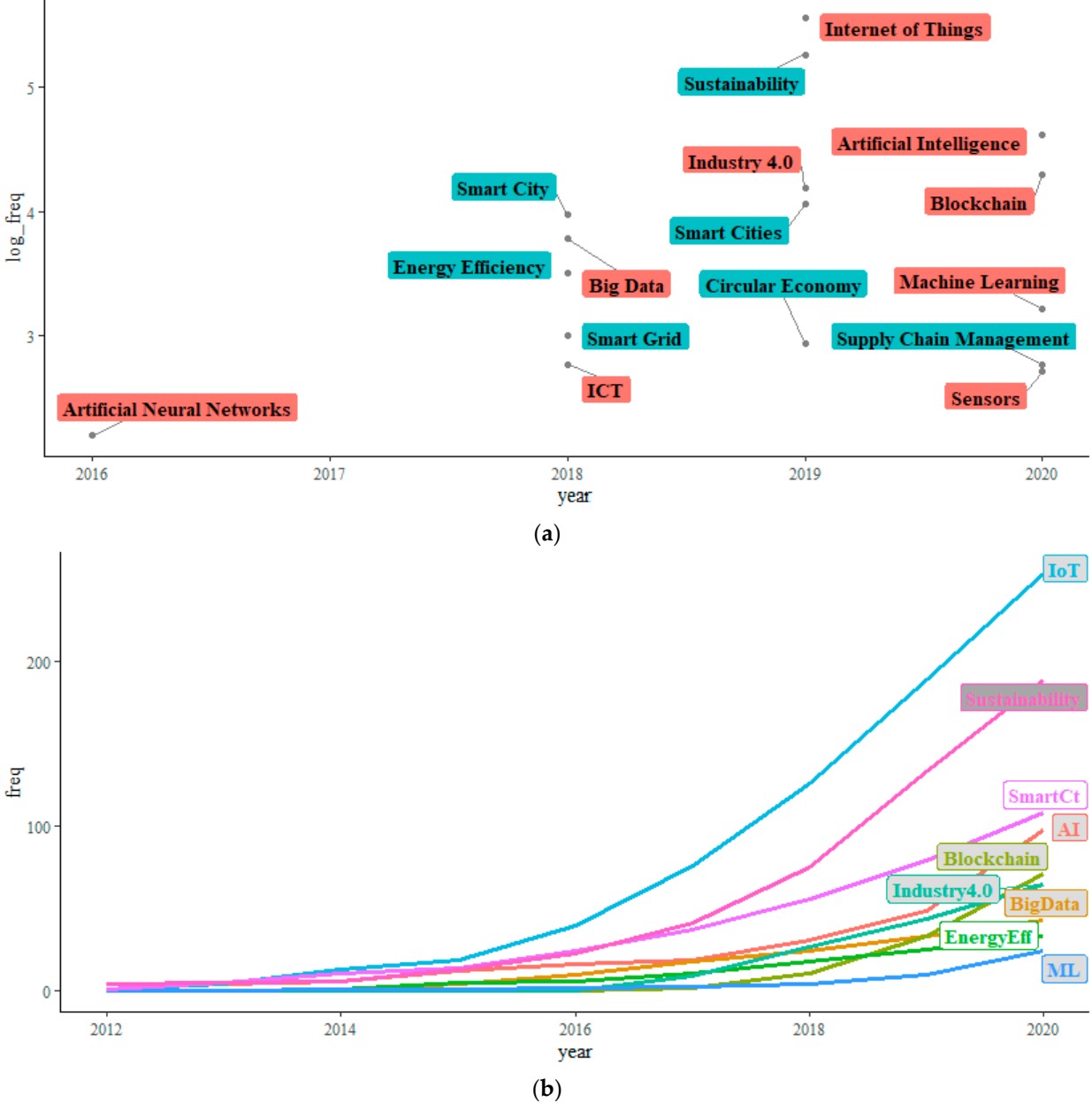

**Figure 5.** Trend topics and word dynamics. (**a**) (top figure) Trend topics identified from publications except reviews (*N* = 828) with an annual minimum word frequency criteria of eight. Label fill colors include blue, indicating applications, and red, indicating methods/technologies. (**b**) (bottom figure) The dynamic trend of the top 10 most frequent terms since 2012 from all publications except reviews (*N* = 828). Here, "smart city" and "smart cities" are combined as "SmartCt"; "EnergyEff" stands for "energy efficiency", and "ML" is "machine learning." Label background colors indicate "sustainability", dark gray, methods/technologies, light-gray, and applications, white.

### 3.1.4. Cooccurrence Network on Initial Publications

Of all the article keywords, we found 171 occurred at least three times (excluding reviews). Using the default setting in Biblioshiny [20], as suggested by Aria and Cuccurullo (2017), with isolated nodes removed, the cooccurrence network of these terms produced five clusters (Figure 6). The first cluster was exclusively related to AI. However, big data (analytics) is not identified under the AI cluster (cluster 1), possibly due to its collaboration with IoT technology, which appeared in the fifth cluster. The second and third clusters each had a relatively small number of nodes focused on smart manufacturing and agriculture, respectively. The fourth and fifth clusters had more nodes, devoted to BC and IoT, respectively, with distinctive application areas (Figure 6, Table 4).

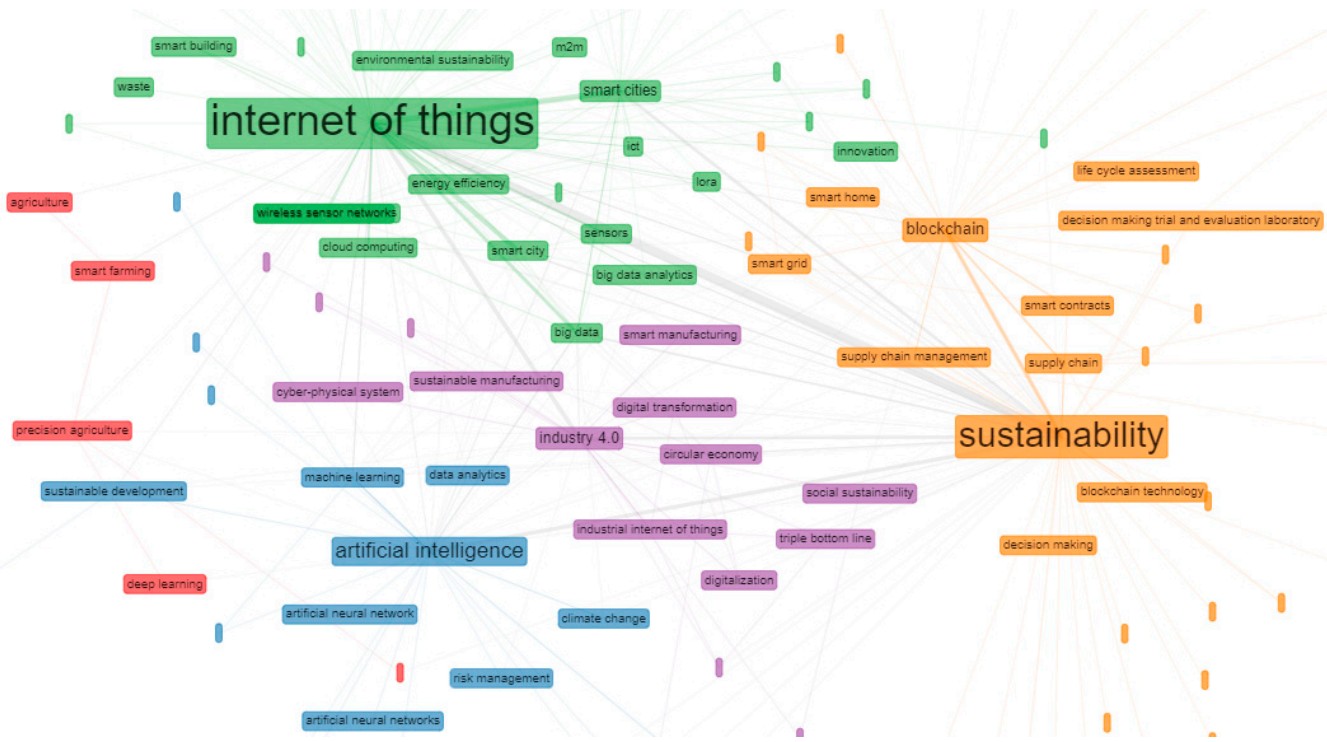

**Figure 6.** A cooccurrence network from all publications except reviews (*N* = 828) with five clusters emerged, building on a total of 171 of the authors' keywords that appeared at least thrice and with isolated nodes removed.

**Table 4.** Co-word clusters from all publications except reviews (*N* = 828) among the top 171 most frequent authors' keywords (at least three occurrences); 131 were non-isolated nodes.

| Cluster | Keywords | | Number of Nodes | Color (Figure 6) |
|---|---|---|---|---|
| | **Methods and Technologies** | **Application Areas** | | |
| Cluster 1 | artificial intelligence, artificial neural network(s), machine learning | climate change, risk management | 21 | blue |
| Cluster 2 | Industry 4.0, cyber-physical system, industrial IoT | smart manufacturing, circular economy | 16 | purple |
| Cluster 3 | deep learning | smart agriculture | 5 | red |
| Cluster 4 | blockchain, smart contracts, life cycle assessment | smart grid, supply chain management | 40 | orange |
| Cluster 5 | Internet of Things (IoT), WSN, cloud computing, Lora, ICT, m2m, sensors, big data analytics | energy efficiency, smart building, smart city/cities, waste | 49 | green |

### 3.2. Key Application Sectors

According to the initial bibliometric analysis, and especially from the results of the trend topics and word dynamic analyses (Figure 5), we identified three key application sectors as substantially adopting these new technologies, including: (i) smart city—city; (ii) energy system—energy; and (iii) supply chain—SC. According to the cooccurrence network, the agriculture and manufacturing sectors also tend to apply these technologies extensively. Nevertheless, they were not identified as trending topics and thus are not included in the sector-specific review. Circular economy (a trending topic in 2019) was not listed as a separate sector, as it implies a general concept and applies to various sectors. Following research workflow Steps D and E, a second round of literature search and screening for each sector was conducted. A total of 285, 189, and 139 publications were identified for city, energy, and supply chain, respectively. Duplications of publications under different sectors were common; for example, 117 out of the 189 publications under "energy" also belonged to "city", indicating that energy systems are a focus area among city topics.

#### 3.2.1. Cooccurrence Network on Each Key Sector

Smart city: Of the authors' keywords observed for the 285 publications related to smart city, 57 terms occurred at least thrice. Four clusters emerged in the cooccurrence network after removing isolated nodes (Figure 7a). IoT received the widest application, covering various topics such as "smart building", "smart mobility", "smart governance", and "waste". Some key technologies of IoT (e.g., cloud computing, sensors) were identified under the IoT cluster, while another three specific technologies of IoT were separately included in the smallest cluster (purple). The other two clusters focused on AI (blue) and BC (green), where AI appeared under the sustainability cluster and BC was related to the smart grid.

Energy: Of the authors' keywords observed for the 189 publications related to energy system (including energy generation, distribution, and consumption), 39 terms occurred at least thrice. Three clusters emerged, with the smallest cluster including only two terms, while the other two focused on IoT and AI (Figure 7b). Similar to smart city, IoT-related technologies received the widest application, covering various topics such as "smart building", "smart energy", "energy management", "energy efficiency", "energy-saving", and "building information modeling". The AI cluster included BC and related to topics such as "renewable energy", "prosumer", and "smart grid". A further context analysis on selected key publications in the next section explains this clustering, as AI was more adapted to the energy generation side (e.g., renewable energy sources) and BC to the energy distribution/market, especially in the smart grid.

Supply chain: Of the authors' keywords observed for the 139 publications related to supply chain, 36 terms occurred at least thrice. Four clusters emerged, with "visibility", "traceability", and "transparency" as some of the keywords identified for the supply chain specifically (Figure 7c). Unlike city and energy, where IoT was identified as the key technology adopted, the key technology applied for supply chain was BC. The smallest cluster only included two terms specifically dedicated to AI, indicating that it received the least and most isolated application to supply chain issues.

A total of 17, 13, and 20 review publications for city, energy, and supply chain, respectively, were separately analyzed, providing readers a brief understanding of each review (e.g., focus areas, the technologies discussed, and information on systematic reviews conducted; see Supplementary Information "SI_ContentAnalysis"). Clearly, IoT was identified as the key technology for cities and energy, while BC is the key technology discussed under supply chain. While the earliest publication mentioned in reviews can be traced back to 1999, most publications included in the reviews appeared in or after 2014.

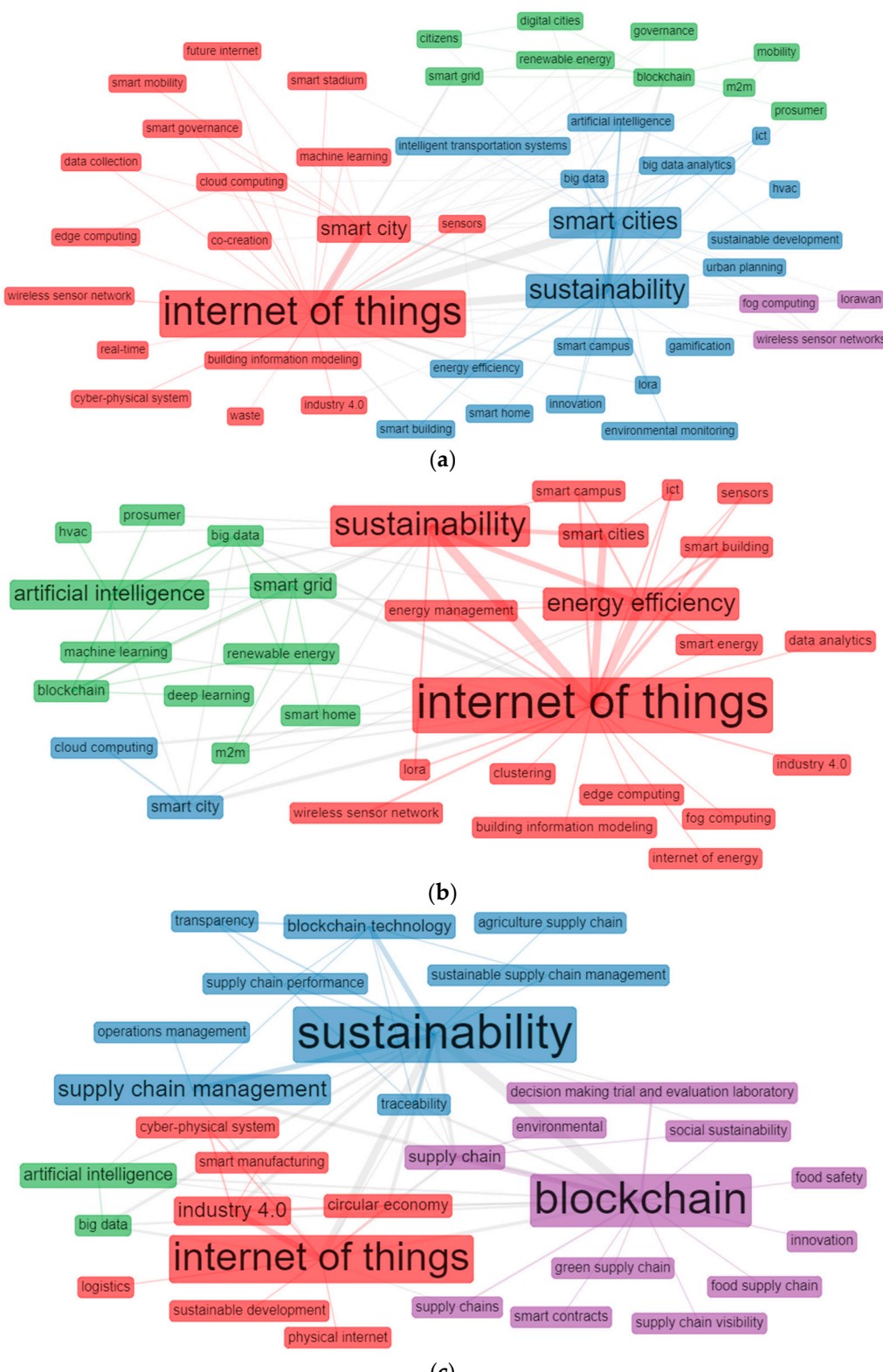

**Figure 7.** Cooccurrence networks for smart city (**a**), energy (**b**), and supply chain (**c**). Each color indicates a clustering theme.

### 3.2.2. Content Analysis on Key Publications

Smart city: As an emerging topic since early 2010, especially in or after 2016 with the establishment of several smart city projects in the European Union (EU), smart city has attracted increased attention among policymakers, researchers, and practitioners. Unfor-

tunately, no standard and shared definition is available yet, and it has various meanings adopted in different contexts [39]. One of the most widely adopted definitions of smart city, offered by Giffinger et al. (2007), is based on six dimensions: smart economy, smart people, smart governance, smart mobility, smart environment, and smart living [40]. It is no surprise that studies aimed at theoretical discussion tended to embrace more dimensions, many of them including all six of these characteristics [41–44]. In contrast, studies that incorporated specific goals and were tested with real-world projects typically targeted only one or just a few dimensions. Among the six dimensions, most studies focused on smart environment and/or smart mobility, while smart economy and smart people were the least mentioned (Table S4).

Technology-wise, IoT was the most discussed real-world tested technology among AI/BC/IoT. Many of the designed IoT platforms incorporated small-scale experimentation. For example, smart campuses have been used as testbeds for smart cities, since major features of a campus resemble those of a city [45]. Fraga-Lamas et al. (2019) assembled a detailed review of various IoT platforms deployed at 16 different smart campuses during 2011–2018 [46]. At the city scale, the Organicity Project was deployed at selected EU cities [47,48]. A pioneer IoT city-scale testbed was also found for the SmartSantander project in the EU [49,50]. More recently, other projects have been established in other OECD countries, such as Australia [51,52], as well as in developing countries [53,54].

AI was most applied to scheduling, predicting/forecasting, and monitoring problems. For example, it is widely applied for environmental (e.g., water, air) monitoring [52]. Recent studies have demonstrated the integration of AI with IoT (Table 5), such as applying big data analytics on IoT sensor data. Some studies involving big data analytics tended to be vague in describing which specific AI models or data processing techniques were being applied [51,55,56]. Compared to IoT and AI, BC received fewer real-world applications, as most BC-related studies provided theoretical discussions and/or proposed conceptual models (Table 5), with the exception of Shojaei et al. (2020), who provided a simulation model where BC was used for the life-cycle management of buildings to improve built asset sustainability [57].

**Table 5.** Example publications for the smart city sector and the technological solutions proposed/adopted. A full list of publications is available in Table S4. The differentiation in the scope of each study (i.e., design/test/simulation/analysis/conceptual model) is also provided in the Excel sheet.

| Smart City Dimensions | |
|---|---|
| Smart economy | [58] |
| Smart people | [59] |
| Smart governance | Participation in decision-making: [47,48]<br>Public and social services: [49,54,60] |
| Smart mobility | Local accessibility: [46,50,61]<br>Sustainable, innovative, and safe transport systems: [51,62] |
| Smart environment | Environmental protection: [52,60,63,64]<br>Attractiveness of natural conditions: [64]<br>Pollution: [47,52]<br>Sustainable resources management: [46,54,65,66] |
| Smart living | [59,67] |
| Studies that did not include real-world applications typically engaged more dimensions and were not included in the above list. | |



**Table 5.** *Cont.*

| Smart City Dimensions | |
|---|---|
| **Technology application** | |
| AI (with real-world case applications) | MAS: [66]<br>ABM: [65]<br>ML: [67]<br>ANN: [60,62]<br>CNN: [68]<br>LSTM: [52,61]<br>Other: [51,55,56,58] |
| AI: fuzzy logic, autonomous systems, SVM, DBN, etc. | [42,69–74] |
| IoT (with real-world case applications) | RFID, QR code/barcode: [47,54]<br>Sensors: [45,46,50,51,56,59,61–64,66,67]<br>Cloud computing: [45,47,50,56,59,63,64,66] |
| BC | Hyperledger fabric: [57]<br>Ethereum: [75]<br>NEO: [74] |
| **Integration of technologies** | |
| AI + IoT | Theoretical discussion and/or conceptual model: [41,42,44,71,72]<br>Designed and tested with real-world cases: [51,52,56,61,62,66–68] |
| AI + BC | Conceptual model: [73] |
| IoT + BC | Conceptual model: [43,75,76] |
| AI + IoT + BC | Conceptual model: [74] |

Energy: For energy systems, most studies focused on energy consumption (e.g., reducing consumption, increasing efficiency) within buildings using IoT. Similar to smart city, IoT was the most applied technology in energy sector. In the few studies focused on energy generation, it applied to renewable energy, where AI was most often applied. At the same time, AI was more devoted to answering prediction-related problems, and BC was the least applied technology, with a niche focus on energy market transactions. Compared to IoT and AI, most BC-related studies remained theoretical discussions instead of real-world applications.

Similar to the city sector, the combination of IoT and AI has gained popularity in recent research. An early example was provided in Uribe et al. (2015), where data collected via IoT technologies (e.g., presence and temperature sensors) were analyzed using fuzzy logic and decision trees for thermal energy management in buildings [77]. Sehovac et al. (2019) applied deep learning models to forecast a building's sensor-based energy load [78]. Among the 11 studies published in 2020, five explicitly applied IoT, including four that integrated data collected by IoT with AI algorithms for various purposes, including: (1) to predict energy consumption using LSTM [79]; (2) to optimize an energy management strategy using a hybrid optimization model [80]; (3) to predict operation processes in manufacturing plants using ANN [81]; and (4) to optimize adaptive power management for grid-connected hybrid renewable energy using ML models [82].

Studies on energy systems have paid more attention to user engagement and feedback than other sectors. Among the 13 works listed in Mataloto et al. (2019) that designed IoT energy management platforms, eight involved user feedback and six explicitly adopted user behavior modeling [83]. The concept of "gamification" was explicitly adopted in several reviewed studies focusing on energy saving in public buildings through user education and increasing awareness [84–86]. Similarly, Mataloto et al. (2020) proposed the "Things2People" concept by using various predictive models (e.g., LSTM) and combining IoT sensor data (Table 6).

**Table 6.** Example publications for the energy sector and the technological solutions proposed/adopted. A full list of publications is available in Table S5. The same definition on the scope of each study (i.e., design/test/simulation/analysis/conceptual model) from "city" was applied.

| Energy Systems | |
| --- | --- |
| Energy generation | [87–89] |
| Energy distribution and market | [90] |
| Energy consumption | [77,79,81,83–86,91–98] |
| Studies that did not include real-world case applications were not included in the above list. | |
| Technology application | |
| AI (with real-world case applications) | Fuzzy logic: [77]<br>Random forest: [99]<br>LSTM: [79]<br>ANN: [81,88,99]<br>Others: [89,94] |
| AI (conceptual model with simulation, RNN, DNN, ML) | [78,80,82,100,101] |
| IoT (with real-world case applications) | RFID, QR code/barcode: [96]<br>Sensors: [77,81,83–87,92,93,98,102] |
| BC (conceptual model) | [90,103–105] |
| Integration of technologies | |
| AI + IoT | Conceptual model and/or simulation: [80,106]<br>Designed and tested with real-world cases: [77,79,81] |
| IoT + BC | Conceptual model: [107] |
| AI + IoT + BC | Conceptual model: [31] |

Supply chain: Eighty out of the one hundred and thirty-nine publications related to supply chain were published in or after 2020, clearly indicating that supply chain is a new trending topic, as shown in Figure 5a. Reviewed publications were not separately listed out because only five publications engaged with real-world case applications (Table 7), all of which adopted IoT (Table S6). While IoT was the most applied technology for the city and energy sectors, BC was the prominent solution for supply chain, primarily because BC has characteristics (e.g., accountability, auditability, transparency, traceability, security) that solve some key concerns in SCM. One of the key topics for supply chain, especially with respect to the food supply chain, was "traceability" [108], a term mentioned in all publications adopting BC into supply chains.

Like smart city and energy, most explorations of BC involved theoretical discussions instead of real-world adoptions, with the exception of some industrial cases [8,109,110]. Building on these case studies, which were led by industry leaders such as Walmart, Maersk, and Alibaba [110], one cohort of studies focused on inductive reasoning. They discussed BC applications, practices, challenges, opportunities, and barriers for general SCM [109–111], the food supply chain [112,113], and the circular economy [8]. Another cohort of studies developed models for investigating BC and identifying enablers, challenges, and barriers for adopting BC in SCM (Table S6). Some questioned the feasibility of BC for SCM due to a high system startup cost [114,115], or its performance "in terms of effectiveness, efficiency, and sustainability" [116]. When testing validation of a proposed BC-based system was reported, it was performed based on expert opinion [115] or in a simulated network environment [117], rather than tested and/or deployed in the real world. In contrast, IoT was more often used in real-world cases when applied to supply chain [118,119].

The trend for technology integration in supply chain was to combine BC and IoT [10], in contrast to the coupling of AI and IoT observed in the city and energy sectors. IoT technologies (e.g., RFID) support data collection, while BC ensures system transparency/immutability. AI was much less applied in supply chain than in the city and energy sectors, except for in

Zhang et al. (2020), suggesting big data analytics and visualization for BC-based life-cycle assessment [115].

**Table 7.** Studies involving real-world cases for each key sector (smart city, energy, supply chain) and the application scale (e.g., from city scale to building/site/company scale). Studies engaging simulations only and/or discussing/evaluating other real-world cases are not listed here.

| | Smart City | Energy | Supply Chain |
|---|---|---|---|
| No. of studies reviewed for content analysis | 38 | 32 | 41 |
| No. of studies engaging real-world cases | 18 | 22 | 5 |
| Case study scale: | | | |
| City (all under column "smart city") | [47–52,60–62] | | |
| Community/campus (all under column "smart city") | [46,54,59,65,66] | | |
| Building (all under column "energy") | [77,79,83,84,86,91–94,96–98,102,120] | | |
| Site (e.g., stadium, watershed, park, lake, river, farm) | [63,64,67] | [87,89] | [121] |
| Company/plant | [58] | [81,88,95,122] | [123,124] |
| Infrastructure | NA | [90] | NA |
| Smart city | NA | NA | [118,119] |

## 4. Discussion

This study sought to examine the current applications of AI/BC/IoT in the context of sustainability and identify the key sectors that have adopted AI/BC/IoT to advance sustainability. The findings of this systematic review indicate that several key industry sectors—smart city, energy, and supply chain—perceive technology and sustainability as a competitive advantage to a sustainable socioecological transition under the current economic development path.

While previous surveys tend to investigate the technologies and their application for a specific sector, we intentionally did not limit our scope to a certain application area or sector. Instead, we provided a comprehensive review from the perspective of the technologies themselves to understand the research trends and industrial applications. Consequently, this approach enabled us to scrutinize our findings through the lens of the UN Environment Programme's SDGs to understand how these new technologies serve in a sustainable socioeconomic transformation and respective environmental implications.

We found that only a few of these goals have been addressed by new technologies. For studies on the smart city, most coincide with Clean Water and Sanitation (Goal 6), Industry, Innovation, and Infrastructure (Goal 9), Sustainable Cities and Communities (Goal 11), and Climate Action (Goal 13). Studies in the energy cohort fit Affordable and Clean Energy (Goal 7) and Sustainable Cities and Communities (Goal 11), and those in supply chain echo Responsible Consumption and Production (Goal 12). Many of the SDGs are rarely discussed, including No Poverty (Goal 1), Quality Education (Goal 4), Gender Equality (Goal 5), Decent Work and Economic Growth (Goal 8), Reduced Inequalities (Goal 10), Life Below Water (Goal 14), Life on Land (Goal 15), and Peace, Justice, and Strong Institutions (Goal 16).

Relatively speaking, a positive outcome cannot be foreseen. A recent study conducted by the World Economic Forum and PwC in 2020 confirmed that over two-thirds of the SDGs could be bolstered by technological innovation [4]. A strong correlation at the national scale between innovation scores and SDG progress was present; however, at the industrial scale, not all aspects of the SDGs received equal attention [4]. Goals 3, 7, 9, and 11 received the highest numbers of new technology applications, while Goals 1, 5, 14, and 15 received the lowest. This finding agrees with ours, and the question remains how government and industries can harness the full benefit of new technologies to achieve more SDG aspects.

### 4.1. Limitation

The database source that we used only included the WoS core collection, while several other previous surveys adopted multiple sources, including Scopus, Google Scholar, etc. We acknowledge that the single database adopted may have led to the exclusion of some relevant articles from the review. Nevertheless, this was a compromise made to smooth the follow-up analysis while ensuring relative inclusiveness. By including 960 publications after an initial screening in the bibliometric analysis, with 38, 32, and 41 studies for each of the key sectors (i.e., SC, energy, SC) in the detailed content analysis, we believe our sample to be representative.

An important aspect to consider, albeit outside the scope of this study, is the technological assessment of AI/BC/IoT. From the furious debate over "cryptocurrency mining" to a recent influx of public interest paid to nonfungible token art and associated "gas fees", the energy consumption of BC has been the subject of a heated debate since its inception. Moreover, additional issues commonly mentioned include scalability. The social justice issues inherent to AI have also led to an ongoing discussion among researchers and policymakers [125,126]. While the technological assessment of AI/BC/IoT, especially from the perspective of sustainability, is of paramount importance in determining their potential applications, this is not within the scope of this study and is left to future research and discussion.

Lastly, while big data analytics, AI, IoT, and BC offer solutions to smart cities and sustainability goals, they also bring limitations in the shape of challenges to privacy (i.e., how to collect, store and analyze data) outside of business economic development purposes (e.g., management, optimization, effectiveness, innovation, productivity, etc.). Moreover, these limitations mean overlooking issues related to the different dimensions of sustainability, which creates a need for novel measures and mechanisms that ensure trustable data acquisition, transmission, and processing in order to guarantee the integrity of services in the context of sustainability [39].

### 4.2. Future Direction

Among the technologies discussed, IoT was the most used in real-world applications for the smart city and energy sectors, whereas AI was extensively applied to energy. The adoption of BC has been relatively slow compared to that of AI/IoT, while showing promise for supply chain applications. BC was compared with centralized systems from the perspective of governance structure, system integration, security, and access, and it is especially promising to address social problems [114]. However, the application of BC remains mostly theoretical and is driven by large corporations with the capability and the resources to implement R&D and pilot projects.

A clear trend observed in the reviewed studies is the integration of IoT with AI and/or BC. As IoT technology matures, it is essential to leverage its full potential stemming from the increasing number of interconnected devices and the volume of data. Indeed, recent studies, like those on or after 2020, have shown special interest in integrating AI, especially big data analytics, with IoT technology. This is a clear trend for both smart cities and energy due to technological advancements such as Hadoop for the efficient storage and processing of big data [36,45]. For supply chain, researchers have proposed that IoT serves data collection, while BC ensures system transparency/immutability [24,114]. Similarly, Sandner et al. (2020) proposed full technological integration to realize the full potential, where IoT is used to collect data, BC to provide infrastructure, and AI to optimize processes [2].

Technology integration, although hyped in theoretical discussions since 2020, has rarely been seen in real-world applications. Several "pilots and early successes" under the UNEP's digital transformation initiatives [3], such as the Coalition for Digital Environmental Sustainability (CODES)—a community "co-creating and accelerating a sustainable digital future for all through a common Action Plan for a Sustainable Planet in the Digital Age" [127]—have demonstrated a positive outlook. Considering that research on AI/BC/IoT applications to advance sustainability has surged since 2016, we are at a critical

stage to integrate these new technologies to create systematic changes along with individual programs. In doing so, this could fulfill the 2030 Agenda for Sustainable Development.

## 5. Conclusions

We reviewed the adoption and applications of AI/BC/IoT in the context of sustainability science. Surging research interest has been observed since 2016 despite notably low collaborations among research institutions. Smart city, energy, and supply chain were identified as the key sectors that have mostly adopted AI/BC/IoT in order to achieve sustainability goals. IoT offers the most real-world applications for smart environment and smart mobility among the six smart city dimensions. IoT was also the most prominent technology adopted by energy industries, especially for reducing energy consumption. AI was less widely adopted compared to IoT but was effective in solving scheduling, predicting, and/or monitoring various problems. Despite the numerous benefits and promises offered by BC, especially in tackling social sustainability issues, it is still in its nascent stage and has had minimal real-world applications due to various factors such as high startup costs. Technology integration between AI and IoT has begun to emerge for applications in smart city and energy systems, while combining BC and IoT has been proposed for the supply chain sector. Finally, our results demonstrate that few of the 17 SDGs, i.e., Goals 6, 7, 9, 11, 12, and 13, have been addressed by new technologies based on our review of the literature. This echoes the findings from the World Economic Forum and PwC in 2020, which confirmed that over two-thirds of the SDGs could benefit from technological innovation [4]. Furthermore, we found SDGs that were rarely addressed, including No Poverty (Goal 1); Quality Education (Goal 4); Gender Equality (Goal 5); Decent Work and Economic Growth (Goal 8); Reduced Inequalities (Goal 10); Life Below Water (Goal 14); Life on Land (Goal 15); and Peace, Justice, and Strong Institutions (Goal 16). To pursue sustainable growth, these SDGs should be considered in the applications of big data analytics, as well as in AI, IoT and BC, with particular consideration given to energy consumption, GHG emissions, and challenges to privacy. Given this evaluation, the challenge remains for government and industries as to how to reap the full benefits of new technologies and achieve a sustainable socioecological transition.

**Supplementary Materials:** The following supporting information can be downloaded at: https://www.mdpi.com/article/10.3390/su14137851/s1, SI_ContentAnalysis.xlsx, README.txt, Table S1: Biliometrix.xlxs, Table S2: Collaboration_Network.csv, Table S3: Source_Impact_top20.csv; Table S4: sust_city.xlxs; Table S5: sust_energy.xlxs; Table S6: sust_SC.xlxs; Table S7: SUST1433_raw_WOS.csv; Table S8: Trend_Topics_MinFreq8_ALL960; Table S9: Word_Dynamics_SCcombined_byrow.csv, Table S10: Annual_Production_fig3; Data S1: PlotFigs.R. Figures 3 and 5 were generated through R script "PlotFigs.R". Figures 4, 6 and 7a–c were generated through the R shiny interface—biblioshiny() using the "Bibliometrix.xlsx", and the "sust_city.xlsx/sust_energy.xlsx/sust_SC.xlsx" as an input data file.

**Author Contributions:** Conceptualization, S.R.W.; methodology, S.R.W. and G.S.; formal analysis, S.R.W. and G.S.; validation, J.C.; investigation, S.R.W. and G.S.; data curation, S.R.W., G.S.; writing—original draft preparation, S.R.W.; writing—review and editing, S.R.W., G.S., I.C. and J.C.; visualization, S.R.W.; funding acquisition, C.S. All authors have read and agreed to the published version of the manuscript.

**Funding:** This study is supported by the National Natural Science Foundation of China (No. 41901264). This material is based upon work supported by the National Science Foundation Graduate Research Fellowship under grant no. (DGE-1848739).

**Institutional Review Board Statement:** Not applicable.

**Informed Consent Statement:** Not applicable.

**Conflicts of Interest:** The authors declare no conflict of interest.

## Abbreviations

| Abbreviation | Meaning |
| --- | --- |
| AI | Artificial Intelligence |
| ABM | Agent-based modeling |
| ANN | artificial neural network(s) |
| BC | Blockchain |
| BLE | Bluetooth Low Energy |
| CNN | Convolutional Neural Network |
| DApps | Decentralized Apps |
| DL | Deep learning |
| DNN | Deep Neural Network |
| DT | Decision trees |
| ELM | Extreme learning machine |
| GA | Genetic algorithms |
| GCS | Global Citation Score |
| GPS | Global Positioning System |
| ICT | Information and Communications Technology |
| IoE | Internet of Everything |
| IoT | Internet of Things |
| LCS | Local Citation Score |
| LoRaWAN | Long Range Wide Area Network |
| LPWAN | Low Power Wide Area Network |
| LSTM | Long short-term memory neural networks |
| LTE | Long Term Evolution wireless broadband |
| M2M | Machine-to-Machine |
| MAS | Multi-agent systems |
| ML | Machine Learning |
| MQTT | Message Queuing Telemetry Transport |
| N2N | Node-to-Node |
| NB-IoT | Narrowband Internet of things |
| NFC | Near Field Communication |
| P2P | Peer-to-Peer |
| PoS | Proof of Stake |
| PoW | Proof of Work |
| RFID | Radio-frequency identification |
| SDGs | Sustainable Development Goals |
| TAK | Title, Abstract, and Keywords |
| UNEP | UN Environment Programme |
| WoS | Web of Science |
| WSN | Wireless Sensor Network |

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
