# Peer review of "A Review on the Adoption of AI, BC, and IoT in Sustainability Research"

_sustainability, doi:10.3390/su14137851_

Round 1
Reviewer 1 Report
The authors of the manuscript sustainability-1746487 entitled “A review on the adoption of AI, BC, and IoT in sustainability research,” conducted a survey to examine the adoption of AI, BC, and IoT techniques in sustainability research in the manuscript. This survey paper presents various types of AI, BC, and IoT algorithms employed in sustainability research. The performance of the given methods in the literature is assessed by a new taxonomy. It offers comprehensive state-of-the-art methods heading to performance evaluation of the given techniques and discusses vital difficulties and possibilities for extensive research. The survey is interesting, extensive, seems feasible, and provide cutting edge research directions. However, the major observations of the reviewer are listed below, which must be addressed before publication.
1. The Abstract section should be concise and informative and must be having flow. Additionally, the abstract is missing some important information. The authors should point out quantitative and qualitative survey metrics. They should add a sentence summarizing the main purpose of the paper.
2. The second part of the introduction enlist few challenges and directions, which are not convincing. Significant challenges related to the topic are missing and cutting edge research directions are needed.
3. This manuscript lacks the connection between the sections and subsections of this survey. Please add more explanations, create connection between sections and subsections, and give solid reasoning why this survey is conducted and which outcomes will bring it.
4. The related work part of the manuscript is very rudimentary, please add more recent and relevant survey papers, and highlight the research gaps in the existing surveys which are already conducted. Besides, compare your survey with some recent surveys and highlight which novelty your survey contributes to the research community.
5. Current limitations, future scope, scalability issues, cutting edge research directions, and also some other useful information should be addressed in the Conclusion section of the survey.
Reviewer 2 Report
Dear author(s),
First of all, thank you for the opportunity to review this manuscript.
I congratulate the authors for the detailed, thorough study. Please, see below some comments and minor issues that can contribute to improve your work.
· The abstract is well written and contains the main elements of the paper. The chosen keywords complement the title and abstract, contributing to the study being found by interested researchers.
· The introduction is well-structured and provides the necessary elements for the reader to understand the research gap and the aim of the authors. The authors show the importance of the theme in an adequate way, citing appropriate references.
· The method is well explained and is suitable for both the proposed objectives and the existing research gap.
· The results are consistent with the literature and the methodology.
· The conclusions are supported by the results.
· Please state the meaning of abbreviations in their first use (e.g., LoRa, MQTT, etc.)
· The resolution of all figures should be improved
· Please consider indicating “a” and “b” in the Figure 3, not only in the its notes; consider the same for Figures 4, 5, 7 (consider also visual formatting)
· “Supply chain” (line 415) needs to be bolded for standardization
Reviewer 3 Report
Dear Authors,
I see that the manuscript is submitted to answer the demands of the special issue: Artificial Intelligence and Big Data Analytics for enhanced Business Operations: A Contemporary Research Framework and Modelling.
As a plus to the mentioned subject, you have additionally talked about BC (blockchain). Also, you emphasize that the BC interest in the next decade will grow and will have an impact of socio-economic tracebility along environmental implications.
As it can be seen, you have approached a qualitative search regarding :
Which venues are addressing the research focus (i.e., application of AI/BC/IoT in the context of sustainability), and what are the research trends, and the leading industries/sectors adopting AI/BC/IoT?
and
How are the leading sectors addressing AI/BC/IoT or adopting them in real-world applications for promoting sustainability?
I appreciate the effort that you put here in order to bring an in depth perspective regarding the proposed questions., however there are some issues that you should also address:
- The BC issue that you have included must be described in terms of it's impact to the environment. The Blockchain works on cryptocurrency which works with cryptomining, and the latter works demand highly efficient technology to mine and high consumption of electricity. I recommend that you introduce a chapter that holds your blockchain support in the future with a sustainability perspective in the future. Keep in mine that it necessary to draw some limitations to this chapter.
- secondly, the conclusion section is too short for your manuscript. Try to bring the highlights of your remarks to the paper.
I hope that you will take into account the above issues in your next manuscript submission.
Good luck with your research.
Round 2
Reviewer 1 Report
I have no further comments
Author Response
Re: "I have no further comments"
Response:
Dear Reviewer 1,
Thank you for your generous time and effort in this review process. We are grateful for your recommendations and believe they significantly improved the quality of this work. We have further edited the English language/grammar throughout the manuscript, per your recommendation.
Best,
Gabriela Shirkey
Reviewer 3 Report
Dear Authors,
I agree that the manuscript has improved. I am glad that you have taken into account my suggestions and that you have highlighted your remarks on the limitations of the study. Also, the extended conclusion has added to paper by underlining intriguing aspects regarding challenges as energy consumption. You have also brought up the subject of the technological implications in the world towards improvement in multiple areas of life.
Thus, even though there are some effects to address to the use of technology as consumption, emissions and others, the use of artificial intelligence, internet of things and their applications as blockchain, will also bring the answers needed towards a sustainable growth.
Author Response
Re: Reviewer 2 comments
"I agree that the manuscript has improved. I am glad that you have taken into account my suggestions and that you have highlighted your remarks on the limitations of the study. Also, the extended conclusion has added to paper by underlining intriguing aspects regarding challenges as energy consumption. You have also brought up the subject of the technological implications in the world towards improvement in multiple areas of life.
Thus, even though there are some effects to address to the use of technology as consumption, emissions and others, the use of artificial intelligence, internet of things and their applications as blockchain, will also bring the answers needed towards a sustainable growth."
Response:
Dear Reviewer 2,
Thank you for your generous time and effort in this review process. We are grateful for your recommendations and believe they significantly improved the quality of this work. We have further edited the English language/grammar throughout the manuscript, per your recommendation.
Best,
Gabriela Shirkey